# Mobile Vaccination Teams for Improving Vaccination Coverage in the Kyrgyz Republic: Results of a National Health System-Strengthening Project during the First Two Years of the COVID-19 Pandemic

**DOI:** 10.3390/children10101681

**Published:** 2023-10-12

**Authors:** Kubanychbek Monolbaev, Alyia Kosbayeva, Marzia Lazzerini

**Affiliations:** 1WHO Country Office, Bishkek 720040, Kyrgyzstan; kmonolbaev@unicef.org; 2United Nations Children’s Fund (UNICEF), 160 Chui Avenue, Bishkek 720040, Kyrgyzstan; 3WHO Regional Office for Europe, DK-2100 Copenhagen, Denmark; kosbayevaa@who.int; 4WHO Collaborating Centre for Maternal and Child Health, Institute for Maternal and Child Health, IRCCS “Burlo Garofolo”, 34137 Trieste, Italy; 5London School of Hygiene & Tropical Medicine, London WC1E 7HT, UK

**Keywords:** vaccination, mobile clinics, COVID-19, children, Kyrgyz Republic, implementation research

## Abstract

(1) Background: This implementation study reports on the results of the mobile vaccination teams’ (MVTs) activities during the first two years of the COVID-19 pandemic in Kyrgyzstan, when other vaccination services were disrupted. (2) Methods: Through a national health system-strengthening project under an order of the Ministry of Health, in 2020, the number of MVTs was increased, focusing on internal immigrant settlements around the cities of Bishkek and Osh and geographically remote areas. MVTs provided free vaccination services. (3) Results: MVTs vaccinated a total of 125,289 and 158,047 children in 2020 and 2021, respectively. The higher contribution of MVTs to vaccination coverage was in children under 5 years of age, with the three top vaccines being IPV (8.9%), MMR (7%), and PCV (6.6%). In 2021, 13,000 children who had not received an IPV vaccination and 8692 children who had not received the Pentavalent vaccine (DPT-HBV-Hib) were reached. The number of cases of vaccine-preventable disease reported in official statistics has reduced over time. (4) Conclusions: MVTs increased vaccination coverage in Kyrgyzstan, in particular in remote regions and migrant settlements, where it accounted for a considerable proportion of the vaccinated. This study adds to previous evidence in the literature of the role of MVTs as a strategy to improve immunization in hard-to-reach populations, particularly children.

## 1. Introduction

Vaccines are among the most cost-effective health interventions [1,2,3]. They are critical to the prevention and control of infectious diseases and their complications, including cancer, and represent a vital tool in the battle against antimicrobial resistance [4]. Recent estimates calculate that 97 million deaths will be avoided between 2000 and 2030 due to vaccinations against ten key pathogens [2]. Moreover, vaccines could prevent 24 million people from falling into poverty by 2030 [3]. Immunization is therefore playing a critical role in achieving many of the Sustainable Development Goals (SDGs), and in particular SDG3—to ensure healthy lives and promote well-being for all at all ages [1]. Vaccines are also an indisputable human right [1].

However, recent estimates point out that, after achieving large gains in vaccine coverage worldwide, in the most recent times in much of the world progresses stalled or even reversed [5]. In a recent Global Burden of Disease Study, out of 204 countries and territories, 94 recorded decreasing combined diphtheria, tetanus toxoid and pertussis (DPT3) coverage since 2010, while only 11 countries/territories reached the national Global Vaccine Action Plan target of at least 90% coverage for all assessed vaccines in 2019 [5].

The World Health Organization (WHO) Immunization Agenda 2030—which is the most recent WHO strategic document on global vaccinations—clearly identifies as strategic priorities both coverage and equity and emphasizes the importance of identifying effective interventions to vaccinate hard-to-reach populations, such as those living in remote areas, and mobile populations [1].

In Kyrgyzstan, a lower-middle-income country of about 6.9 million people, recent surveys showed large disparities across regions and settings in immunization rates [6,7,8]. Previous assessments showed that there are two major groups of hard-to-reach populations in terms of vaccination [6,7,8]. The first group is composed of families living in geographically remote areas: about 90% of Kyrgyzstan’s territory is located 1500 m above sea level, and 66% of its inhabitants live in rural settings, with seasonal migration to mountainous regions for summer pastures still being a widespread phenomenon [6]. The second group is composed of internal migrants, who move from rural areas to the major cities, in particular Bishkek and Osh, seeking employment opportunities and better living conditions [6,7]. According to the most recent data, which may however underestimate more-recent figures, internal migrants accounted for 48.5% of the population in Bishkek and 28.5% of Osh’s population [6].

In both the above-described groups, vaccination rates have been reported as unsatisfactory, in particular due to delays in completing vaccinations in children when compared to those according to the national calendar [6]. One of the key reasons identified for low vaccination rates was low access to health care and distance from health services [6,8]. Poor vaccine education and growing vaccine hesitancy have also been reported in Kyrgyzstan, in particular in the rural population [6].

Mobile vaccination teams (MVTs) have been successfully used in the past to vaccinate hard-to-reach populations such as, but not limited to, nomadic and seminomadic populations in Africa and the Middle East [9,10]. In Kyrgyzstan, MVTs were established in 2018, with the support of a national health system-strengthening project, funded by the Global Alliance for Vaccine Immunization (GAVI), under the technical coordination of the WHO Country Office, with the main objective of improving access to immunization services for hard-to-reach populations.

However, the advent of the COVID-19 pandemic imposed additional constraints on the implementation of immunization services in the country. From March 2020, in Kyrgyzstan, routine vaccinations in Primary Health Care (PHC) facilities were suspended with an order of the Ministry of Health (MOH)—order no. 207 dated 29 March 2020—for two months [11]. Later, COVID-19 affected the delivery of essential health services in many ways, including decreasing the number of staff available to work [11]. The COVID-19 pandemic caused 51% of the households in Kyrgyzstan to lose income—with the top causes being the inability to work or losing a job—and it reduced access to health care by up to 38% in some areas, such as Osh Region [12].

In this context, in May 2020, it was decided to further escalate MVTs’ activities in Kyrgyzstan. This aimed to mitigate the disruption to the immunization services and the expected related consequences, such as accumulation of susceptible individuals, a higher likelihood of outbreaks, and an increased burden on health systems already strained by the response to the COVID-19 outbreak. This paper aims to assess key results of the MVTs’ activities in Kyrgyzstan during the first two years of the COVID-19 pandemic.

## 2. Materials and Methods

### 2.1. Study Design

This paper summarizes the results of a national health system-strengthening project. The study design is implementation research and is reported according to the Standards for Reporting Implementation Studies (StaRI) statement [13]. More details are reported in Appendix A.

### 2.2. Setting

In Kyrgyzstan, the government is responsible for vaccination provision to all citizens free of charge [6]. The national laws imply ensuring provision of free vaccination to all Kyrgyz citizens at any location in the country, irrespective of migration status [6].

Most vaccinations are provided at PHC, consisting of three levels of facilities: (1) at the lower level, there are 1030 Feldsher-Obstetrical Ambulatory Points, which are the main providers of PHC in rural areas; (2) at the middle level, there are 696 Family Group Practices, which usually consist of three to five doctors; (3) at the upper level, there are 64 Family Medicine Centers and 28 General Practice Centers, which represent the largest outpatient health facility types, employing several doctors and providing outpatient services including some diagnostics and minor surgeries [6]. Additionally, the provision of immunization services through private health facilities including private maternity facilities is allowed by national legislation.

The national immunization schedule is summarized below [6] and provided in a tabular format in Appendix A.

The Bacillus Calmette–Guérin (BCG) vaccine and the first dose of the Hepatitis B (HBV) vaccine are given soon after birth, at hospital level.Children receive within the age of 12 months vaccinations with 3 doses of Pentavalent (containing Diphtheria, Pertussis, Tetanus, HBV, and Heamophilus influenza type B (Hib B)), 3 doses of Oral Polio (OPV), one dose of Inactivated Polio Vaccine (IPV), 3 doses of Pneumococcal 13-valent vaccine (PCV), the first dose of Measles, Mumps, and Rubella (MMR); 3 doses of the Rotavirus 5-valent vaccine (RV).Another DPT dose is given at 24 months. The MMR 2nd dose and a booster of Diphtheria and Tetanus [DT] is given at 6 years of age; other boosters of the Tetanus and Diphtheria [Td] vaccine are given at 11 and 16 years and every 10 years thereafter.Epidemiological surveillance of infectious, including vaccine-preventable, diseases and parasitic diseases is conducted by the Public Health Services (Department and Centers of Disease Prevention and State Sanitary epidemiological surveillance—DP and SSES). Each case of these infections is registered and notified to the Centers of DP and SSES. Statistics are freely accessible (http://www.stat.kg/en/opendata/category/260/ (accessed on 25 September 2023)). According to national statistics, in 2019, 2380 cases of measles, 1340 cases of epidemic parotitis, 436 cases of pertussis, and no cases of diphtheria were reported.

### 2.3. Implementation of MVTs

The national project was implemented with an order of the MOH of the Kyrgyz Republic. The MVTs focused their activities on hard-to-reach areas, identified by a list developed by the Kyrgyz MOH, and consisting of internal immigrant settlements around the cities of Bishkek and Osh and geographically remote areas in seven other regions (for a detailed list, see Appendix A).

Since May 2020, the number of MVTs gradually increased from 43 to 65 teams by the end of 2020 and to 78 teams by the end of 2021. Mobile vaccination rounds were increased from four rounds per year to six rounds per year during both 2020 and 2021. Number of MVTs varied in each region according to local needs. Detailed dates for each round are provided in Appendix A.

Each MVT consisted of a doctor and a vaccination nurse, as per the national regulation. They were nominated by the PHC facilities among staff already responsible for vaccinations (i.e., mostly pediatricians, family doctors, and vaccination nurses). All team members were recruited among local staff, serving the geographical territory where they were already working. The project provided financial support for transport and per diem. Technical coordination was performed by staff of the WHO Country Office in dialogue with relevant health authorities.

The team travelled mostly in a hired car. In selected cases, horses were used to reach families of shepherds in high mountains where there was no road. National measures on infection prevention and control were followed during mobile immunization sessions in the field. The MVT staff received training in infection prevention and control measures, including wearing of personal protective equipment.

Other interventions implemented in the country during the time of MVT implementation included a vaccine information campaign, coordinated by the Republican Health Promotion Centre and Red Crescent Society, and other similar vaccine demand increase activities (e.g., community awareness) organized by UNICEF.

### 2.4. Data Collection and Monitoring

Each doctor in an MVT, according to pre-defined national procedures, which included pre-defined data collection forms, collected data on the vaccinations administered. Data linkage with the personal identity of each person vaccinated was performed through the unique identification number (UIN), which in Kyrgyzstan is assigned to each individual, including Kyrgyz citizens, resident non-citizens, and stateless persons [14]. Kyrgyzstan operates a unified population registration system designed to combine independently developed digitized systems for civil registration, registration of address of residence, and issuance of ID cards and travel documents [14]. The UIN is automatically assigned when a birth is registered and linked to all subsequent vital events recorded under that person’s name. A UIN also ensures that a person’s vital events can only be registered once [14].

Each MVT was under the coordination of the respective regional immunologist. In addition, data monitoring and data analysis were conducted after each round of vaccination by two national coordinators working at the National Center for Immunoprophylaxis. Data were checked for completeness, correctness of calculation, and internal consistency. Additionally, during on-site monitoring visits, the achievements of the MVTs were discussed and random checks were performed on individual reports of the MVTs.

### 2.5. Data Analysis

We analyzed the following key indicators of MVTs activities: (1) the rate of identified hard-to-reach settlements visited by MVTs (calculated as the number of settlements visited by the MVTs out of the list of settlements identified by the MOH); (2) the absolute number of people vaccinated by MVTs, total and by each vaccine, in 2020 and 2021; (3) the contribution of MVTs to the total immunization coverage—the latter being defined as the proportion of vaccinated out of the total population requiring a vaccine (calculated based on health service census performed twice a year by health facility staff in their catchment area), in the country and in each region, in 2021; (4) the absolute number of children who were reached by MVTs in 2021, among those who had missed vaccinations identified as national priority (IPV and Penta); (5) trends over time (2019 vs. 2020 vs. 2021) of reported cases of vaccine-preventable diseases (for national statistics, see http://www.stat.kg/en/opendata/category/260/ (accessed on 25 September 2023)).

Data are presented as absolute numbers and frequencies with respective proportions for categorical parameters. We also reported narratively on the key constraints encountered in implementing MVTs, and key determinants of success, in the views of the coordinators (K.M., A.K.).

## 3. Results

### 3.1. Rate of Identified Hard-to-Reach Settlements Visited by MVTs

MVTs covered 86% and 88% of the hard-to-reach settlements, as identified by the MOH list. Coverage varied across regions and in each round, with some remote areas, such as Jalal-Abad region, being more difficult to cover, while others, such as Talas and Chui Region, where the MVT was about 100% of the settlements in each round, were easier. More details on coverage by region and MVT rounds are reported in Appendix A.

### 3.2. Number of Children and Adults Vaccinated

In 2020, a total of 165,478 people were vaccinated during six MVT rounds, including 90,483 children under 1 year, 125,289 children under 16 years of age and 18,949 adults. Just in the period from May to December 2020, MVTs vaccinated a total of 155,003 people, including 80,153 children under 1 year of age and 18,949 adults. Details by region are reported in Appendix A.

In 2021, a total of 171,161 people were vaccinated during six MVT rounds, including 109,267 children under 1 year of age, 158,047 children and teenagers under 16 years of age, and 13,114 adults. Details by region are reported in Appendix A. When compared to 2020, in the year 2021, there was a significant increase in the number of people vaccinated for each single vaccine, except for Td (Figure 1).

### 3.3. Contribution of MVTs to the Total Immunization Coverage in the Country

MVTs increased vaccination coverage in the country, with a contribution to the total immunization coverage variable by vaccine, age group, and region (Figure 2 and Figure 3, Appendix A).

By age, overall, the highest contribution of MVTs was observed in children under 5 years of age, and in this class, the three vaccines where MVTs brought the largest contribution to coverage were IPV (8.9%), MMR (7%), and PCV (6.6%) (Figure 2A). In the older age group, the contribution of MVTs to vaccination coverage was less variable, with the top two vaccines being MMR at 6 years and Td at 6 years (both 5.8%), followed by Td at all other age groups (Figure 2B).

When data were further analyzed by region, in both age groups, the highest MVT contribution was observed in three regions: Chui Region (24.4% for IPV, 14.2% and 13.4% for Td at 6 and 11 years, respectively), Talas Region (21.3% and 19.5% for MMR at 6 and 11 years, respectively, 21.2% for Td at 6 years, 19.3% for PCV), and migrant settlements around Osh city (16.9% for IPV) (Figure 3 and Figure 4).

### 3.4. Children Who Had Missed Vaccinations and Who Were Reached by MVTs

MVTs succeeded in reaching a total of 13,000 children who had not received IPV vaccination and 8692 children who had not received Penta-3 (DPT-VGV-Hib) (Table 1); out of these, 6271 (28.9%) were vaccinated in Chui Region.

### 3.5. Trends over Time of Reported Cases of Vaccine-Preventable Diseases

According to national statistics (http://www.stat.kg/en/opendata/category/260/ (accessed on 25 September 2023), the number of reported cases of measles reduced over time from 2380 (2019) to 733 (2020) to 6 (2021). Cases of epidemic parotitis reduced from 1340 (2019) to 120 (2020) to 66 (2021). Cases of pertussis reduced from 436 (2019) to 97 (2020) to 5 (2021). No cases of diphtheria were reported in the study period.

### 3.6. Constraints and Determinants of Success

Three key constraints were observed in relation to MVTs’ activities: (1) difficulties in accessing very remote areas, characterized by lack of roads and large distance between settlements, resulting in increasing time needed to visit the settlement; (2) conflict in border areas between Kyrgyzstan and Tajikistan, reducing feasibility of MVTs in these areas; (3) lack of staff due to COVID-19 infection, and drain on human resources due to being allocated to COVID-19-related health services.

The following key aspects were considered as major determinants for success: (1) joint planning with relevant departments of the MOH; (2) availability of adequate financial resources from the GAVI HSS2 grant; (3) regular implementation of the MVT rounds; (4) adaptation of MVT rounds to hard-to-reach settlements based on coverage data and mobility of the population; (5) flexibility of MVTs to vaccinate population when and where they were available in hard-to-reach settlements.

## 4. Discussion

This report shows that MVTs significantly contributed to vaccination coverage in Kyrgyzstan, in particular for children under 5 years of age and in selected regions, such as Chui and Talas, and migrant settlements around Osh city, where it accounted for a considerable proportion of the total population vaccinated. Despite the COVID-19 pandemic, the number of people vaccinated by MVTs increased over time, in parallel with the number of mobile teams enrolled, thus suggesting a dose–effect response. Differences on the MVTs contribution by region and vaccine can be justified by several factors, including: the rate of settlements visited by MVTs (higher for Talas and Chui Region, where MVTs had the higher impact); the vaccination needs in each region (variable over time based also on the previous rounds of vaccination and new births); the population acceptance for vaccines. The decreasing number of cases for many vaccine-preventable diseases over the study period suggests a potential role of MVTs, although other factors (e.g., vaccine-awareness campaigns) may have contributed.

This study adds to evidence in the literature of the role of MVTs as a strategy to improve immunizations, in particular for hard-to-reach populations and among children. There are relatively few studies on the effectiveness of MVTs. Two previous studies among the nomadic population in Chad and Bedouin minorities in Israel demonstrated a beneficial effect on vaccination rates after the implementation of MVTs [9,10]. Another study reported increased coverage of the influenza vaccine among health workers in Spain after the implementation of MVTs [15]. Recently, health authorities in different settings, including Canada, Singapore, and India, adopted MVTs as one of the strategies to roll out COVID-19 vaccination [16,17,18]. MVTs have also been recommended by a systematic review to improve vaccination rates in Ethiopia [19].

In most of these previous studies, MVTs have been used in the context of complex intervention tailored to the specific setting. Interventions implemented together with MVTs included, for example, informative campaigns, delivered in different ways including by health professionals [10], dedicated telephone lines [14], posters [14], emails [14], dedicated websites [14]; offering economic incentives and benefits, such as scholarships for training [10] or prizes for vaccinated individuals [14]; social intervention such as creating a positive stigma [10,14], involving local leaders/influencers [9,14]; and other marketing strategies such as a webpage entitled “I’ve already been vaccinated”, showing photos of vaccinated people [14].

Other cross-cutting interventions used in combination with MVTs or in the context of interventions to improve the vaccination rate among nomadic populations included improving vital registration and case tracking [10]; increasing human resources dedicated to vaccination [10]; building capacities for mobile immunization services [9,10]; recruiting and training local community health workers [19,20]; reorganizing health services to increase access to vaccination services [10]; intersectoral collaboration [10].

Nevertheless, even in previous reports on MVTs, several constraints to achieve the expected vaccination rate have been reported, with the most frequent ones being: budget constraints [9,10]; difficulties in recruiting doctors and nurses, and in consider economic incentive to attract public-health staff to work in remote area [10]; difficulties in identifying all eligible individuals due to the lack of vital registration or to internal movements of the population [9].

In the literature, there is a lack of head-to-head intervention studies enabling understanding of which component of a vaccination strategy—for example, if either the implementation of MVTs or other components such as economic incentives—is more effective in expanding vaccination coverage to hard-to-reach populations [9,10,14,17]. On the other hand, it is very plausible that the effectiveness of different components is affected by many factors, including (1) the context and setting where they are implemented; (2) how each component is actually delivered (for example, the effectiveness of an informative campaign may change drastically based on what key message is prioritized and how it is communicated, what material is developed, and how it is used, such as what influencer/pictures are chosen and so on); (3) how each component is integrated in the routine health system, ensuring internal coordination, access to resources, and sustainability over time.

As such, we believe that the findings of this study may be of use to other countries, in particular for reaching nomadic populations, but they are not directly generalizable to other settings. In each setting, the most effective implementation strategies need to be carefully tailored to the local context, based on existing needs.

The literature offers some very good examples of studies performing formative research, such as evidence synthesis, before deciding what specific interventions to adopt [9]. However, not all projects aiming at improving vaccination coverage reported a proper assessment of the underlying causes for low vaccination rates [10,14,17,21] or a theory of change for the interventions proposed. Additionally, most studies on vaccine interventions did not involve end-users in their design. We believe that vaccination strategies should be designed based as much as possible on the context-specific evidence of underlying causes for low vaccination rates, together with end-users and health professionals involved in vaccination delivery. A relevant example of user-centered design to develop interventions related to vaccination has been documented for Papillomavirus and COVID-19 [22,23]. As additional resources, we recommend (1) the WHO Guide to Tailoring Immunization Programmes (TIP), which provides support to national immunization programmes in designing vaccination responses that are setting and context specific [24]; (2) the European Centre for Disease Prevention and Control Catalogue of Interventions Addressing Vaccine Hesitancy, which includes a list of 40 interventions to improve vaccination coverage [25].

We acknowledge as possible limitations of this study that we did not report other relevant outcomes, such as disease incidence, whether MVTs also affected vaccine education, vaccine hesitancy, or costs of the intervention. However, it must be considered that all other studies on MVTs limited their reporting on vaccination coverage and the number of people vaccinated [9,10,14]. Disease incidence, in particular for some conditions such as measles, mumps, rubella, rotavirus, and pneumococcal infections, may be affected by several biases, including underreporting and lack of diagnostic facilities or even changing of diagnostic criteria [14]. Measuring accurately and reliably vaccine education is complex and also open to reporting bias. Documenting vaccine hesitancy deserves separate studies that explore the many factors—including women empowerment, vaccine literacy, risk perception, lack of trust, fear of adverse effects and pain—that, according to the literature, are associated with it [26,27,28,29,30]. A detailed analysis of the costs of the intervention was beyond the scope of this report but may be the focus of future publications.

## Figures and Tables

**Figure 1 children-10-01681-f001:**
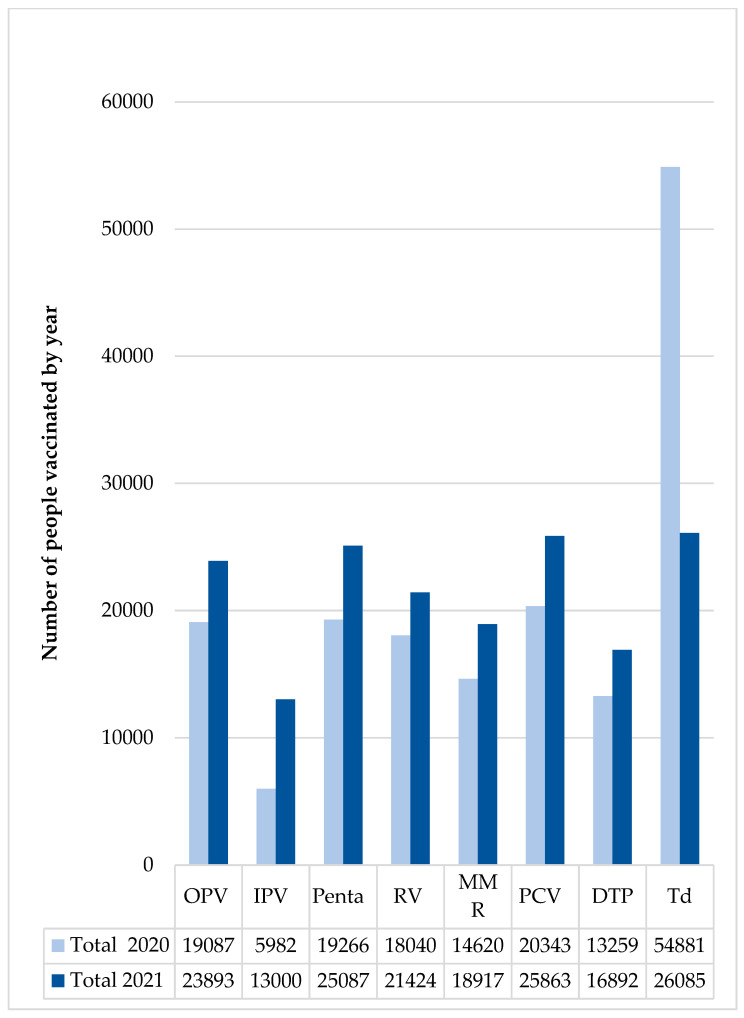
Number of people vaccinated in 2020 and 2021. Abbreviations: DPT = diphtheria, pertussis, tetanus vaccine; IPV = inactivated polio vaccine; MMR = measles, mumps, rubella vaccine; OPV = oral polio vaccine; Penta = pentavalent vaccine; PCV = pneumococcal vaccine; RV = rotavirus vaccine; Td = tetanus diphtheria vaccine.

**Figure 2 children-10-01681-f002:**
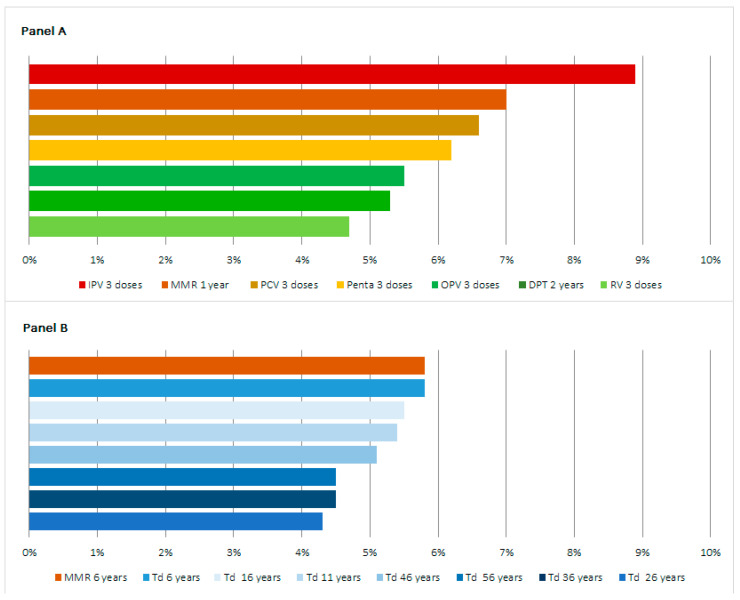
Contribution of MVTs to total vaccine coverage in children under 5 years (**A**) and over 5 years of age (**B**), 2021. Abbreviations: DPT = diphtheria, pertussis, tetanus vaccine; IPV = inactivated polio vaccine; MMR = measles, mumps, rubella vaccine; OPV = oral polio vaccine; Penta = pentavalent vaccine; PCV = pneumococcal vaccine; RV = rotavirus vaccine; Td = tetanus diphtheria vaccine.

**Figure 3 children-10-01681-f003:**
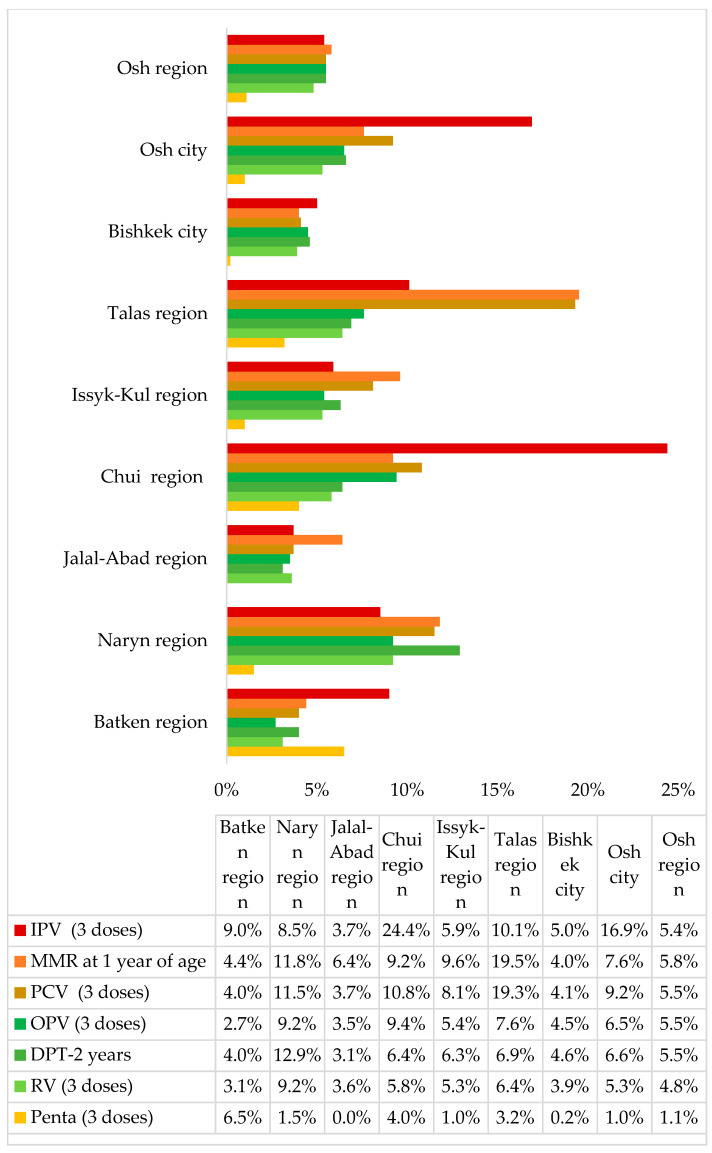
Contribution of MVTs to total vaccine coverage, by region, for vaccines in children under 5 years of age, 2021. Abbreviations: DPT = diphtheria, pertussis, tetanus vaccine; IPV = inactivated polio vaccine; MMR = measles, mumps, rubella vaccine; OPV = oral polio vaccine; Penta = pentavalent vaccine; PCV = pneumococcal vaccine; RV = rotavirus vaccine; Td = tetanus diphtheria vaccine.

**Figure 4 children-10-01681-f004:**
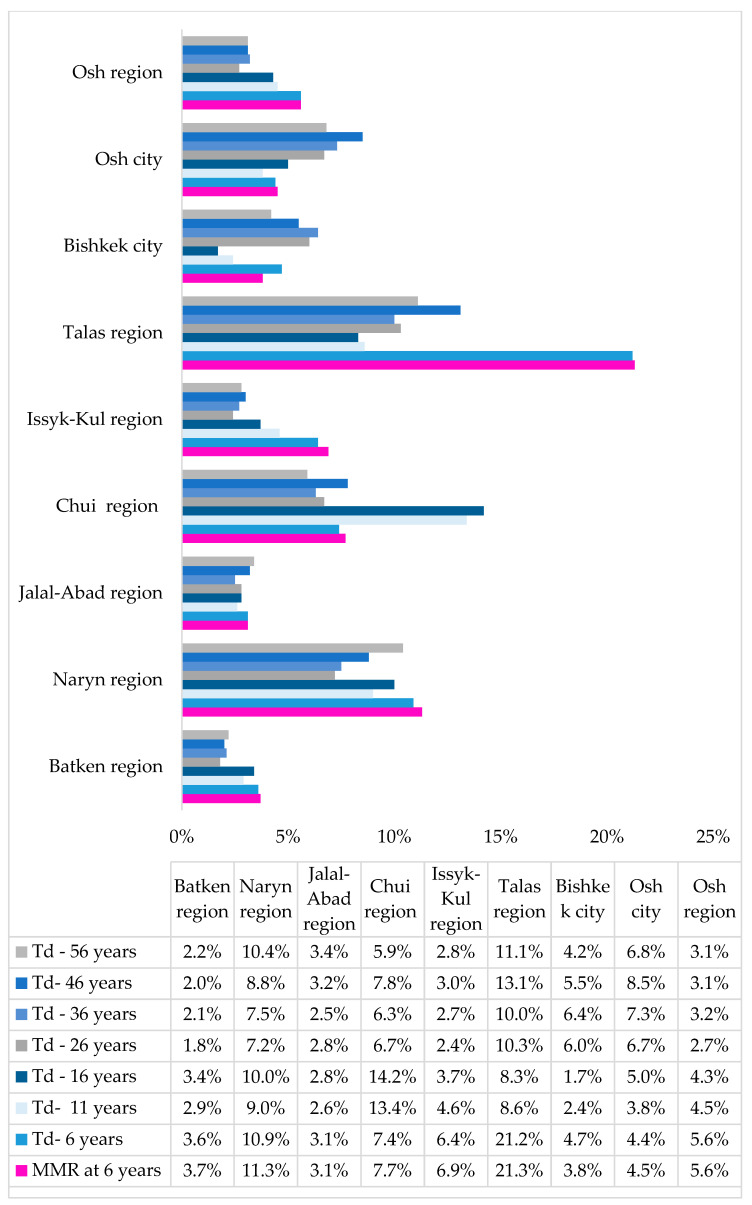
Contribution of MVTs to total vaccine coverage, by region, for vaccines in the population of over 5 years of age, 2021. Abbreviations: MMR = measles, mumps, rubella vaccine; Td = tetanus diphtheria vaccine.

**Table 1 children-10-01681-t001:** Children who had missed priority vaccinations and who were reached by MVTs, 2021.

Vaccinations	Batken Region	Naryn Region	Jalal-Abad Region	Chui Region	Issyk-Kul Region	Talas Region	Bishkek City	Osh City	Osh Region	Total for the Republic
IPV (inactivated polio)	1171	454	1078	4945	531	513	1145	1442	1721	13,000
Penta-3 (DPT-HBV-Hib)	418	500	1814	1926	464	357	893	550	1770	8692

Note: IPV was selected as key indicator because eradicating polio is a national priority, and previous reports feature gaps in polio coverage. Coverage with the penta vaccine was selected because it is a key indicator for the WHO/UNICEF and because it contains 5 antigens.

## Data Availability

Data are contained within the article and Appendix A. Individual level data are not publicly available for privacy reasons.

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
