# Peer review of "Mobile Vaccination Teams for Improving Vaccination Coverage in the Kyrgyz Republic: Results of a National Health System-Strengthening Project during the First Two Years of the COVID-19 Pandemic"

_children, 2023, doi:10.3390/children10101681_

Round 1
Reviewer 1 Report
Very impactful study. I have incredible respect for the team for engaging in this work and studying it systematically.
There is an 8 in line 171 that appears to be an error.
In Figure 1, the vaccine name should appear in one line-otherwise looks confusing.
I don't think the authors answer the question of if these children would have been vaccinated by another means if this program did not exist. This should be made clear--whether this is an actual increase or just shift in vaccine delivery model.
Author Response
Dear editors,
We resubmit after the 1st round of reviews the paper titled “Mobile Vaccination Teams for Improving Vaccination Coverage in the Kyrgyz Republic: Results of a National Health System-Strengthening Project during the First Two Years of the COVID-19 Pandemic”
Please find a point-by-point answers to referees’ comments below.
__________________________________________________________
REVIEWER #1
Very impactful study. I have incredible respect for the team for engaging in this work and studying it systematically.
- There is an 8 in line 171 that appears to be an error.
*** Thank you, this has been corrected
- In Figure 1, the vaccine name should appear in one line-otherwise looks confusing.
*** Thank you, this has been corrected
- I don't think the authors answer the question of if these children would have been vaccinated by another means if this program did not exist. This should be made clear--whether this is an actual increase or just shift in vaccine delivery model.
*** thank you, we have made this point clearer. We have also added Trends over time of reported cases of vaccine preventable diseases.
REVIEWER #2
Dear author,
This article examines mobile vaccination teams for improving vaccination coverage in the Kyrgyz Republic in the first two years of Covid-19 Pandemic. The author discusses the methods and effects of mobile vaccination to health systems and vaccination coverage. It's evident that the authors and all of the healthcare workers in this work have dedicated considerable effort to this vaccination program. However, in some sections, more detailed information could be provided. So I have a few comments and questions about this manuscript
- It would be helpful if you could add your national vaccination schedule in a table form organized by months.
***This has been added in the Supplementary files. A text description was provided in page 3 of the manuscript, and it was optimised
- Could you provide some information regarding the prevalence of vaccine-preventable diseases in your country such as measles, diphtheria, pneumococcal infections and your surveillance system?
***We have added this info in the introduction section of the paper. Epidemiological surveillance of infectious (including vaccine-preventable diseases) and parasitic diseases is conducted by the Public Health Services (Department and Centers of Disease Prevention and State Sanitary-epidemiological surveillance - DP and SSES). Each case of these infections ais registered and notified to the Centers of DP and SSES. Department of DP and SSES has a database for all these notifiable infections.
Link to data http://www.stat.kg/en/opendata/category/260/
- Which type of pneumococcal vaccine is being administered in your country? (13, 8 ?)
***PCV13, this info has been added in the paper in page 3, and in APPENDIX table A2
- In some countries, the rotavirus vaccine is not included in national schedule due to economic reasons. It's a great opportunity that you can provide this vaccine. How many doses and which type of rotavirus vaccine do you administer
***RV-5 (Rotavirus 5-valent) vaccine - three dose, this info has been added in the paper in page 3, and in APPENDIX table A2s
Reviewer 2 Report
Dear author,
This article examines mobile vaccination teams for improving vaccination coverage in the Kyrgyz Republic in the first two years of Covid-19 Pandemic. The author discusses the methods and effects of mobile vaccination to health systems and vaccination coverage. It's evident that the authors and all of the healthcare workers in this work have dedicated considerable effort to this vaccination program. However in some sections , more detailed information could be provided.So I have a few comments and questions about this manuscript
1. It would be helpful if you could add your national vaccination schedule in a table form organized by months.
2. Could you provide some information regarding the prevalence of vaccine-preventable diseases in your country such as measles, diphtheria, pneumococcal infections and your surveillance system ?
3.Which type of pneumococcal vaccine is being administered in your country? (13, 8 ?)
4. In some countries, the rotavirus vaccine is not included in national schedule due to economic reasons. It's a great oppurtinty that you can provide this vaccine. How many doses and which type of rotavirus vaccine do you administer ?
Author Response
- It would be helpful if you could add your national vaccination schedule in a table form organized by months.
***This has been added in the Supplementary files. A text description was provided in page 3 of the manuscript, and it was optimised
- Could you provide some information regarding the prevalence of vaccine-preventable diseases in your country such as measles, diphtheria, pneumococcal infections and your surveillance system?
***We have added this info in the introduction section of the paper. Epidemiological surveillance of infectious (including vaccine-preventable diseases) and parasitic diseases is conducted by the Public Health Services (Department and Centers of Disease Prevention and State Sanitary-epidemiological surveillance - DP and SSES). Each case of these infections ais registered and notified to the Centers of DP and SSES. Department of DP and SSES has a database for all these notifiable infections.
Link to data http://www.stat.kg/en/opendata/category/260/
- Which type of pneumococcal vaccine is being administered in your country? (13, 8 ?)
***PCV13, this info has been added in the paper in page 3, and in APPENDIX table A2
- In some countries, the rotavirus vaccine is not included in national schedule due to economic reasons. It's a great opportunity that you can provide this vaccine. How many doses and which type of rotavirus vaccine do you administer
***RV-5 (Rotavirus 5-valent) vaccine - three dose, this info has been added in the paper in page 3, and in APPENDIX table A2s
Round 2
Reviewer 2 Report
Revised manuscript has been assessed, author's detailed revision is acceptable.